# Effect of Modulator Therapies on Nutritional Risk Index in Adults with Cystic Fibrosis: A Prospective Cohort Study

**DOI:** 10.3390/nu16121811

**Published:** 2024-06-08

**Authors:** Nadir Yalçın, Esen Deniz Akman, Oğuz Karcıoğlu, Karel Allegaert, Kutay Demirkan, Ebru Damadoğlu, Ali Fuat Kalyoncu

**Affiliations:** 1Department of Clinical Pharmacy, Faculty of Pharmacy, Hacettepe University, 06800 Ankara, Türkiye; nadir.yalcin@hacettepe.edu.tr (N.Y.); esendeniiz@gmail.com (E.D.A.); kutay@hacettepe.edu.tr (K.D.); 2Department of Pulmonary Medicine, Faculty of Medicine, Hacettepe University, 06800 Ankara, Türkiye; oguzkarcioglu@hacettepe.edu.tr (O.K.); edamadoglu@yahoo.co.uk (E.D.); fuattoraks@gmail.com (A.F.K.); 3Department of Pharmaceutical and Pharmacological Sciences, KU Leuven, 3000 Leuven, Belgium; 4Department of Development and Regeneration, KU Leuven, 3000 Leuven, Belgium; 5Child and Youth Institute, KU Leuven, 3000 Leuven, Belgium; 6Department of Clinical Pharmacy, Erasmus MC, 3015 GD Rotterdam, The Netherlands

**Keywords:** cystic fibrosis, adults, nutritional risk index, nutritional therapy, modulators, rare disease

## Abstract

*Background*: Modulator therapies improve weight and body mass index (BMI) in cystic fibrosis (CF) patients. We aimed to compare the nutritional risk index (NRI) in adult CF patients receiving modulator (MT) or only non-modulator (conventional) therapies (non-MT). *Methods*: A single-center prospective cohort study was conducted between June and December 2023. The NRI based on weight gain and albumin was calculated at beginning and end of a 12-week period in both groups. This design was pragmatic, since it was based on individual patient access to MT for 12 weeks. *Results*: In total, 107 patients were included [mean (SD) age: 23.85 (4.98) years, 54.7% male, 46.7% MT]. In the MT group, mean (SD) weight (kg) and albumin (g/dL) increased significantly [changes: +3.09 (2.74) and +0.17 (0.37); *p* < 0.001]. In the non-MT group, weight and albumin decreased significantly [changes: −0.99 (1.73) and −0.12 (0.30); *p* < 0.001]. Compared to the MT group, baseline mean (SD) NRI in the non-MT group was significantly higher [100.65 (11.80) vs. 104.10 (10.10); *p* = 0.044]. At the end of the 12 weeks, mean (SD) NRI in the MT group was higher than in the non-MT group [104.18 (10.40) vs. 102.58 (12.39); *p* = 0.145]. In the MT group, the NRI category improved in 22 (44%), and worsened in 3 (6%) patients (*p* < 0.001). In the non-MT group, the NRI category improved in 2 (3.5%), and worsened in 10 (17.5%) patients (*p* < 0.001). *Conclusions*: This is the first study reporting on a positive effect of MT on NRIs, based on weight gain and albumin. Personalized nutrition and routine follow-up of adults with CF based on NRI is recommended prior to MT initiation.

## 1. Background

Cystic fibrosis (CF) is an autosomal recessive disease caused by a mutation in the gene encoding the CF transmembrane regulator (CFTR) protein. Mutations of the CFTR protein in the lungs result in thick mucus and infections. The CFTR protein is also found in the pancreas. This can lead to pancreatic insufficiency, inflammation, dysmotility, and malabsorption [1]. 

Patients with CF typically have a lower body weight than peer-healthy individuals. Pancreatic exocrine insufficiency leading to intestinal malabsorption and decreased food intake, along with increased energy expenditure during infections, contribute to difficulties in maintaining body weight. A low body mass index (BMI) in CF is associated with impaired lung function and increased mortality. Therefore, guidelines from the Cystic Fibrosis Foundation recommend maintaining a BMI of ≥22 kg/m^2^ in adult women and ≥23 kg/m^2^ in adult men [2].

In malnourished adolescents and adults with cystic fibrosis, conventional strategies of oral dietary supplements and dietary counseling are not known to cause a significant change in energy intake or ideal body weight percentage [3]. Despite the use of pancreatic enzyme replacement therapy, the frequency of malnutrition in CF suggests that aggressive nutritional support and highly effective CFTR modulator therapy are essential for a healthy BMI [4,5,6].

Weight gain after modulator therapy has been evaluated in several studies. In a phase 3 study of elexacaftor/tezacaftor/ivacaftor in F508del heterozygotes, the absolute change from baseline in body weight after 24 weeks of treatment was +3.4 kg compared to +0.5 kg for placebo, with a +2.9 kg difference between groups (95% CI, 2.3 to 3.4) [7]. In patients receiving ivacaftor for G551D mutations, the mean (standard deviation, SD) weight significantly increased from 52.1 (20.4) kg to 54.2 (20.8) kg after 3 months (*p* < 0.001) [8]. The albumin level was also significantly increased after one year in patients treated with lumacaftor/ivacaftor (*p* < 0.001) [9]. 

The Nutritional Risk Index (NRI), originally developed for surgical patients [10,11], has been utilized both in medical and surgical inpatients [12] as well as outpatients with heart failure [13]. It has also been applied at baseline and after transplantation in lung transplant candidates, including patients with or without CF. The NRI, a screening tool rarely utilized in chronic lung diseases previously, is a simple instrument encompassing albumin, weight, and the ratio of ideal body weight (IBW), which can only be detected through medical records and has the potential to stratify nutritional risk based on an individual’s score [14]. Combining both anthropometric and biochemical parameters allows for a more comprehensive assessment of nutritional status, indirectly evaluating calorie intake through assessing factors known to influence albumin levels, such as systemic inflammation, and kidney and liver function [12,15].

The NRI was calculated for patients using the following formula: (15.19 × serum albumin [g/dL]) + (41.7 × weight [kg])/IBW [kg] [16]. Patients were grouped according to the NRI categories of malnourishment: non-malnourished (>100), mild (97.6–100), moderate (83.5–97.5), and severe malnourishment (<83.5) [17]. IBW was used instead of usual body weight because it is less subjective [13]. The aim of this study was to compare the NRI in adults with CF access to modulator treatment for 12 weeks in addition to conventional therapies, or without access to modulator treatment, and to analyze the change in NRI between both groups during this 12-week period.

## 2. Methods

### 2.1. Study Design

This prospective single-center cohort study was conducted from June to December 2023. Patients with CF over 18 years old, followed by the pulmonary medicine outpatient clinic of a tertiary care university hospital, were included. Patients who declined to provide written consent, those who had undergone lung or liver transplantation, and individuals who had been hospitalized within the preceding month were excluded from the study. Due to the fact that modulator therapies are not reimbursed in Turkey, and individual access to treatment is currently available for only three-month time intervals, based on a court decision [18], patients were evaluated 12 weeks (3 months) after the start of the modulator treatment. To enable comparison, CF patients without modulator therapy were also assessed at 12-week intervals. The study was approved by the Local Ethics Committee.

### 2.2. Data Collection

Data were prospectively collected by a clinical pharmacist routinely attending a pulmonary medicine outpatient clinic where the largest number of adults with CF are followed up nationally. Age, gender, genotype, presence of pancreatic insufficiency, CF-related diabetes mellitus and liver disease, use of oral nutritional supplements, pancreatic enzyme replacement, and type of modulator therapies were determined as demographic data of patients who had started and not started modulator treatment according to the results of genetic analysis. Subsequently, albumin, weight, BMI, and NRI scores were obtained simultaneously from the modulator and non-modulator therapy groups at the beginning and end of the 12-week period to assess nutritional risk. 

As a result of the decision made by the responsible physicians of the patients based on the gene tests and the clinical condition of the patient, independent of the study and based on international treatment guidelines, some of the patients continued to receive conventional therapies, while modulator therapies could be added to these therapies for some patients based on the above-mentioned court decision. Therefore, the data obtained in the study completely reflect real-world data. The primary endpoint of the study was to determine the effect of modulatory and non-modulatory treatment on NRI and related parameters at the end of the 12-week period.

### 2.3. Statistical Analysis

It was planned to include at least a total of 70 patients, with 35 patients each in the modulator and non-modulator groups, with an effect size of 0.80, 95% power, and 5% margin of error within the foreseen study period. The calculation of sample size was made with the G-power 3.0.10 program. 

Using descriptive statistics, mean and SD or median and range for numerical variables and number and percentage values for categorical variables were given. The normality assumption was analyzed by the Kolmogorov–Smirnov test. In the comparison of numerical data, the Mann–Whitney U test was used for non-normally distributed data. The Pearson’s chi-squared test was used to compare independent groups in terms of categorical variables. These univariate analyses identified demographic and NRI variables with *p*-values below 0.20. A logistic regression was subsequently performed to determine their effect on the likelihood of changing treatment type (modulator vs. non-modulator). All statistical tests were applied with a 95% confidence interval and the statistical significance level was accepted as *p* < 0.05. All analyses were carried out in the IBM SPSS Statistics Version 23 software.

## 3. Results

### 3.1. Demographic and Clinical Characteristics

A total of 107 patients were included [mean (SD) age: 23.85 (4.98) years, 54.7% male, 46.7% with modulator therapy]. All demographic data of the patients were similar between both groups (*p* > 0.05), except for pancreatic insufficiency and pancreatic enzyme replacement therapy (*p* < 0.05) (Table 1).

### 3.2. Nutritional Risk Index

In the modulator therapy group, mean (SD) albumin (g/dL), weight (kg), BMI (kg/m^2^), and NRI score significantly increased (4.08 [0.44] vs. 4.25 [0.39], 54.65 [13.05] vs. 57.74 [12.85], 19.88 [3.66] vs. 21.02 [3.53], and 100.65 [11.80] vs. 104.18 [10.40], respectively; *p* < 0.001). On the contrary, in the non-modulator therapy group, mean (SD) albumin (g/dL), weight (kg), BMI (kg/m^2^), and NRI score significantly decreased (4.26 [0.48] vs. 4.13 [0.60], 57.97 [10.62] vs. 56.98 [11.28], 20.72 [2.95] vs. 20.36 [3.19], and 104.10 [10.10] vs. 102.58 [12.39], respectively; *p* < 0.001). The weight, BMI, albumin, and NRI values of groups at the beginning and end of the 12-week period are summarized in Table 2. 

At baseline, and at the end of the 12-week period, changes in the modulator and non-modulator therapy groups were detected in weight (IQR 1.00, 4.50 and IQR 1.00, 4.50), BMI (IQR 0.38, 1.79 and IQR 0.38, 1.79), albumin (IQR −0.005, 0.39 and IQR −0.005, 0.39), and NRI score (0.52, 8.32 and 0.52, 8.32), respectively (Table 3).

NRI scores were categorized after calculation. The number of patients without a malnutrition risk increased from 24 to 35 patients in the MT group, while it decreased from 38 to 34 in the non-MT group. In addition, the number of patients with severe malnutrition decreased in the modulator therapy group, while it increased in the non-modulatory therapy group (Table 4).

In the modulator therapy group, the NRI category improved in 22 (44%) patients and worsened in 3 (6%) patients (*p* < 0.001). On the contrary, the NRI category improved in 2 (3.5%) patients and worsened in 10 (17.5%) patients in the non-modulator therapy group (*p* < 0.001) (Table 4). When comparing elexacaftor/tezacaftor/ivacaftor (ETI) to modulator therapies, the baseline mean (SD) was albumin 4.03 (0.46) g/dL vs. 4.22 (0.38) g/dL (*p* = 0.237), weight 51.62 (11.43) kg vs. 65.36 (13.28) kg (*p* = 0.001), BMI 19.20 (3.54) kg/m^2^ vs. 22.28 (3.16) kg/m^2^ (*p* = 0.12), and NRI 98.60 (11.98) vs. 106.64 (9.14) (*p* = 0.044), respectively. At the end of the 12 weeks, the mean (SD) for ETI vs. other therapies was albumin 4.24 (0.41) g/dL vs. 4.31 (0.34) g/dL (*p* = 0.535), weight 54.77 (10.87) kg vs. 68.27 (14.27) kg (*p* = 0.004), BMI 20.37 (3.29) kg/m^2^ vs. 23.31 (3.51) kg/m^2^ (*p* = 0.016), and NRI 103.95 (9.85) vs. 109.76 (9.11) (*p* = 0.066). When elexacaftor/tezacaftor/ivacaftor was compared to other modulator therapies, no significant difference was observed for albumin, weight, BMI, or NRI score (*p* > 0.05).

The logistic regression model was statistically significant. The model explained 74% (Nagelkerke R^2^) of the variance in therapy (modulator vs. non-modulator) and correctly classified 86% of cases. However, there was no significant relationship between the demographic and NRI variables and the treatment type.

## 4. Discussion

This study has shown that while albumin, weight, BMI, and NRI scores were higher in the group not receiving MT treatment compared to the group that did receive MT at baseline, the situation was reversed at the end of the 12-week period. These parameters showed a statistically significant increase in favor of the MT group. In this context, the NRI is a useful and effective tool to assess the risk of malnutrition in adults with CF. A retrospective study from Turkey also showed that children who were not eligible for CFTR modulators had lower median height z-scores and median BMI z-scores than those who were [19]. 

Nutritional screening tools have been developed for children and adolescents with CF [20,21]. To the best of our knowledge, a nutritional screening tool has not yet been validated in adult patients with CF. On the other hand, Schonenberger and Tóth used the Nutritional Risk Score (NRS-2002) in their studies on adult patients [22,23]. In our study, we used the NRI and showed that there was a significant improvement in the group receiving MT compared to the group not receiving MT. 

While it is known that BMI and dietary intake increase with ETI therapy [1], it has not yet been evaluated for malnutrition risk before and after modulator therapies. The current study described the risk of malnutrition over time and its relationship to other variables in patients with and without modulator therapies.

In the study of Sheikh et al., the increase in BMI was 4.4% in a 3-month period with 48 patients (*p* < 0.001), similar to our study [24]. In the study of Bianu et al. examining metabolic changes after lumacaftor treatment with 12 adult patients, albumin level non-significantly increased to 0.2 ± 0.7 after 3 months (*p* = 0.168) contrary to our study (*p* < 0.001) [25]. The large number of adult CF patients in our study may explain this situation. NRI increased after 12 weeks because albumin and weight data were included in the NRI formula. While the malnutrition risk categories of patients using modulator therapy improved, the categories of patients in the non-modulator therapy group worsened significantly (*p* < 0.001). Although weight, albumin, and NRI values were significantly higher in the non-modulator therapy group than in the modulator therapy group at baseline, this difference decreased 12 weeks after treatment with higher values in the modulator therapy group.

Drugs with a higher potency for CFTR correction appear to be associated with a greater weight gain [26]. In this study, when ETI and other modulator therapies were compared, no significant difference was found in the increase in weight and albumin over time and the decrease in the risk of malnutrition. This may be due to the fact that the number of patients receiving ETIs was higher than the others and the albumin and weight values of patients receiving other modulator therapies were higher than those receiving ETIs at baseline.

It has not been evaluated in this study, but in previous studies, decreased lung function and anthropometric measurements were also associated with the risk of malnutrition [27].

This study had several strengths and limitations. There was a significant difference between the two groups in terms of pancreatic enzyme use and pancreatic insufficiency at baseline. Patients could not be followed up in terms of pancreatic insufficiency after modulator therapy, and patients may have stopped using enzymes due to improvements in gastrointestinal symptoms. Recommending the intake of modulator therapy with fatty foods in order to increase their absorption may also increase calorie intake. 

One of the strengths of the study is that according to 2021 European Cystic Fibrosis Society Patient Registry data, there are 352 patients aged ≥ 18 in Turkey [28] and most of these patients are followed by our center and involved in this study. We think that our study population represents adult CF patients in Turkey. 

## 5. Conclusions

The prevalence of malnutrition is high in CF patients. Modulator therapies improve albumin, weight, and BMI based on NRI, but not all patients can access the treatments. NRI is a useful tool to detect malnutrition risk and assist with the prescription or evaluation of the efficacy of medical and nutrition interventions in adult CF patients. However, the suggestion of routine NRI-based follow-ups requires further justification. Future studies could investigate the long-term effects of modulator therapies on NRI and clinical outcomes to further explore the validity of this recommendation.

## Figures and Tables

**Table 1 nutrients-16-01811-t001:** Demographic data of patients.

Variables	Modulator Therapy Group (*n* = 50)	Non-Modulator Therapy Group (*n* = 57)	*p* Value
Gender, *n* (%)			
Female	26 (52)	23 (40.4)	0.228 ^b^
Male	24 (48)	34 (59.6)	
Age (years), mean (SD)	24.12 (5.08)	23.71 (4.91)	0.680 ^a^
Genotype			
f508del/f508del	18 (36)	4 (7)	<0.001 ^b^
f508del/other	10 (20)	12 (21.1)	
Other/other	22 (44)	35 (61.4)	
Unknown	0	6 (10.5)	
Pancreatic insufficiency, *n* (%)	40 (80)	54 (94.7)	0.020 ^b^
CF-related diabetes mellitus, *n* (%)	9 (18)	7 (12.3)	0.408 ^b^
CF-related liver disease, *n* (%)	10 (20)	14 (24.6)	0.572 ^b^
Oral nutritional supplements, *n* (%)	23 (46)	30 (52.6)	0.494 ^b^
Pancreatic enzyme replacement, *n* (%)	40 (80)	54 (94.7)	0.020 ^b^
Modulator therapy, *n* (%)		-	
Elexacaftor/tezacaftor/ivacaftor (ETI)	39 (78)		
Tezacaftor/ivacaftor	1 (2)		
Ivacaftor	10 (20)		

CF: cystic fibrosis. ^a^ Independent *t* test, ^b^ Pearson’s chi-squared test.

**Table 2 nutrients-16-01811-t002:** Variables at baseline at the end of the 12-week period in both groups.

Variables	Modulator Therapy Group (*n* = 50)	Non-Modulator Therapy Group (*n* = 57)	*p* Value
Albumin (g/dL), mean (SD)			
At the beginning of the 12-week period	4.08 (0.44)	4.26 (0.48)	0.029
At the end of the 12-week period	4.25 (0.39)	4.13 (0.60)	0.410
Weight (kg), mean (SD)			
At the beginning of the 12-week period	54.65 (13.05)	57.97 (10.62)	0.049
At the end of the 12-week period	57.74 (12.85)	56.98 (11.28)	0.918
Body mass index (kg/m^2^), mean (SD)			
At the beginning of the 12-week period	19.88 (3.66)	20.72 (2.95)	0.121
At the end of the 12-week period	21.02 (3.53)	20.36 (3.19)	0.349
Nutritional risk index, mean (SD)			
At the beginning of the 12-week period	100.65 (11.80)	104.10 (10.10)	0.044
At the end of the 12-week perio	104.18 (10.40)	102.58 (12.39)	0.145

**Table 3 nutrients-16-01811-t003:** Changes detected in patients over the 12-week follow-up period with and without modulator therapy.

Variables	Modulator Therapy Group (*n* = 50)	Non-Modulator Therapy Group (*n* = 57)	*p* Value
Albumin (g/dL)	0.17 (0.37) −0.63 (−1.24)	−0.12 (0.30) −1.24 (−0.46)	*p* < 0.001
Weight (kg)	3.09 (2.74) −1 (−10.50)	−0.99 (1.73) −8 (−4)	*p* < 0.001
Body mass index (kg/m^2^)	1.14 (0.97) −0.36 (−4.26)	−0.36 (0.69) −3.13 (–1.29)	*p* < 0.001
Nutritional risk index	4.86 (6.72) −8.90 (−27.40)	−2.67 (5.07) −21.40 (−7)	*p* < 0.001

Values are expressed as mean (standard deviation) and minimum–maximum.

**Table 4 nutrients-16-01811-t004:** NRI categories at baseline and after 12 weeks in both groups.

NRI Categories	Modulator Therapy Group (*n* = 50)	Non-Modulator Therapy Group (*n* = 57)
Baseline, *n* (%)		
Non-malnourished (>100)	24 (48)	38 (66.7)
Mild (97.6–100)	7 (14)	4 (7)
Moderate (83.5–97.5)	14 (28)	14 (24.6)
Severe malnourishment (<83.5)	5 (10)	1 (1.8)
After 12 weeks, *n* (%)		
Non-malnourished (>100)	35 (70)	34 (59.6)
Mild (97.6–100)	8 (16)	4 (7)
Moderate (83.5–97.5)	6 (12)	15 (26.3)
Severe malnourishment (<83.5)	1 (2)	4 (7)

## Data Availability

The data presented in this study are available on request from the corresponding author. The data are not publicly available due to restrictions on privacy and ethics.

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
