# Peer review of "Effect of Modulator Therapies on Nutritional Risk Index in Adults with Cystic Fibrosis: A Prospective Cohort Study"

_nutrients, 2024, doi:10.3390/nu16121811_

Round 1
Reviewer 1 Report
Comments and Suggestions for Authors
This cohort study aimed to compare the NRI in adults with CF access to modulator treatment for 12 weeks in addition to conventional therapies, or without access to modulator treatment, and to analyse the change in NRI between both groups during this 12-weeks period.
The introduction consists of a list of definitions and formulae, but reader is given no motivation for the study or any idea of the studies that have taken place before this one, or the efficacy of the conventional therapies that currently exist. This needs to be addressed.
The design is described as a cohort study but contains two groups those receiving modular treatment and those not. No randomisation is mentioned, so it is unclear how the groups were recruited. This is a major weakness in the paper. No hypotheses or primary endpoints are mentioned. The sample size calculation is vague and reports a confidence interval which is inconsistent with the two-group nature of the study. The authors report the mean(sd) comparisons between groups at the start for albumin, weight, BMI and NRI and then write p<0.001 once at the end, which is meaningless. In three of the four outcomes there is a significant difference at baseline which suggests that there may be other non-measured confounding factors. The authors report the mean(sd) comparisons between groups at the end for albumin, weight, BMI and NRI and then write p<0.001 once at the end, which is again meaningless. The same information is then reported in a table which makes the preceding text redundant. The appropriate analysis for this study is a regression of each outcome at week twelve on the baseline outcome values a dichotomous term representing group and as many causal confounders as measured.
The opening line of the discussion should start along the lines with, “This study has shown……”, and then compare the results in this study with previous ones. This will attempt to account for the baseline imbalances.
The conclusion “Modulator therapies reduce the risk of malnutrition” is vague and qualitative and was never shown specifically shown in the paper as malnutrition was never defined or used in the analysis.
Author Response
This cohort study aimed to compare the NRI in adults with CF access to modulator treatment for 12 weeks in addition to conventional therapies, or without access to modulator treatment, and to analyse the change in NRI between both groups during this 12-weeks period.
Thank you for your valuable comments, we confirm.
The introduction consists of a list of definitions and formulae, but reader is given no motivation for the study or any idea of the studies that have taken place before this one, or the efficacy of the conventional therapies that currently exist. This needs to be addressed.
These sentences and references have been added, in line with your request:
In malnourished adolescents and adults with cystic fibrosis, conventional strategies of oral dietary supplements and dietary counselling are not known to cause a significant change in energy intake or ideal body weight percentage (3). Despite the use of pancreatic enzyme replacement therapy, the frequency of malnutrition in CF suggests that aggressive nutritional support and highly effective CFTR modulator therapy are essential for healthy BMI (4, 5, 6).
The design is described as a cohort study but contains two groups those receiving modular treatment and those not. No randomisation is mentioned, so it is unclear how the groups were recruited. This is a major weakness in the paper.
Thank you for pointing this out. No randomisation method was used in the study, as the study was pragmatic in its design in a clinical cohort. In our country, besides the cost of CFTR modulators, another problem is that they are not eligible for all patients with CF due to age and genotype. As a result of the decision made by the physician independently of the study according to the gene tests and the clinical condition of the patient, some of the patients continued to receive conventional treatment, while CFTR was added to the treatment in some of them, following individual legal decisions. Therefore, the data obtained in the study completely reflect real world data. In this context, this paragraph has been added to clarify the issue:
As a result of the decision made by the responsible physicians of the patients based on the gene tests and the clinical condition of the patient, independent of the study and based on international treatment guidelines, some of the patients continued to receive conventional therapies, while modulator therapies could be provided to some of them. Therefore, the data obtained in the study completely reflect real world data.
No hypotheses or primary endpoints are mentioned. The sample size calculation is vague and reports a confidence interval which is inconsistent with the two-group nature of the study.
This sentence has been added to the method section:
The primary endpoint of the study was to determine the effect of modulatory and non-modulatory treatment on NRI and related parameters at the end of the 12-week period. The study was pragmatic, describing the impact of modulator therapy in a clinical cohort, with access to modulator therapy in a portion of these patients.
The authors report the mean(sd) comparisons between groups at the start for albumin, weight, BMI and NRI and then write p<0.001 once at the end, which is meaningless. In three of the four outcomes there is a significant difference at baseline which suggests that there may be other non-measured confounding factors.
In the study, four different analyses were performed at the beginning of the study and at the end of 12 weeks, both within and between groups. The changes in Table 3 are given because it was initially taken into consideration that albumin, weight, BMI and NRI may vary in both groups and that there may be confounding factors both within and between groups. Therefore, the data in Table 2 actually constitute the basis of the data in Table 3. We hope my explanation has clarified this issue.
The authors report the mean(sd) comparisons between groups at the end for albumin, weight, BMI and NRI and then write p<0.001 once at the end, which is again meaningless. The same information is then reported in a table which makes the preceding text redundant. The appropriate analysis for this study is a regression of each outcome at week twelve on the baseline outcome values a dichotomous term representing group and as many causal confounders as measured.
Thank you for your very important suggestions. After critical statistical analyses and to improve the reporting, these sentences have been added to the method and result sections:
Methods: These univariate analyses identified demographic and NRI variables with p-values below 0.20, and a logistic regression was performed to determine their effect on the likelihood of changing treatment type (modulator vs. non-modulator).
Results: The logistic regression model was statistically significant. The model explained 74% (Nagelkerke R2) of the variance in therapy (modulator vs. non-modulator) and correctly classified 86% of cases. However, there was no significant relationship between the demographic and NRI variables and the treatment type.
The opening line of the discussion should start along the lines with, “This study has shown……”, and then compare the results in this study with previous ones. This will attempt to account for the baseline imbalances.
The first paragraph of the discussion section has been added as you requested and an example from the limited literature has been given:
This study has shown that while albumin, weight, BMI and NRI scores were higher in the group not receiving MT treatment compared to the group that will receive MT at baseline, the situation was reversed at the end of 12-week period and these parameters showed a statistically significant increase in favor of MT group. In this context, NRI is a useful and effective tool to assess the risk of malnutrition in adults with CF. A retrospective study from Turkey also showed that children who were not eligible for CFTR modulators had lower median height z-scores and median BMI z-scores than those who were (20).
The conclusion “Modulator therapies reduce the risk of malnutrition” is vague and qualitative and was never shown specifically shown in the paper as malnutrition was never defined or used in the analysis.
Thank you for pointing this out. We agree with you. For this reason, the relevant sentence has been edited:
Modulator therapies improved the albumin, weight and BMI based on NRI in patient co-treated with modulator therapies.

Reviewer 2 Report
Comments and Suggestions for Authors
In this manuscript, the authors presented the effect of modulator therapies on nutritional risk Index in adults with cystic fibrosis with a prospective cohort study. They compare the nutritional risk index (NRI) in adult CF patients receiving modulator (MT) or only non-modulator (conventional) therapies (non-MT) by conducting a single-center prospective cohort study. They firstly reported a positive effect of MT on NRI, based on weight gain and albumin, as well as personalized nutrition and routine follow-up of adults with CF based on NRI is recommended prior to MT initiation. This manuscript is interesting, and the experiments are well thought out and executed.
Author Response
In this manuscript, the authors presented the effect of modulator therapies on nutritional risk Index in adults with cystic fibrosis with a prospective cohort study. They compare the nutritional risk index (NRI) in adult CF patients receiving modulator (MT) or only non-modulator (conventional) therapies (non-MT) by conducting a single-center prospective cohort study. They firstly reported a positive effect of MT on NRI, based on weight gain and albumin, as well as personalized nutrition and routine follow-up of adults with CF based on NRI is recommended prior to MT initiation. This manuscript is interesting, and the experiments are well thought out and executed.
Thank you for your valuable comments, and your overall very supportive assessment.

Reviewer 3 Report
Comments and Suggestions for Authors
The manuscript "Effect of Modulator Therapies on Nutritional Risk Index in Adult Cystic Fibrosis Patients: A Prospective Cohort Study”, presents a prospective cohort study comparing the nutritional risk index (NRI) in adult cystic fibrosis (CF) patients receiving modulator therapies (MT) versus those receiving only non-modulator (conventional) therapies (non-MT). The study assesses weight gain and albumin levels over a 12-week period in both groups.
Comments:
- The study design as a single-center prospective cohort study is appropriate for assessing the impact of MT on NRI.
- The manuscript mentions a pragmatic design based on individual patient access to MT for 12 weeks, which could potentially introduce bias. Providing more details on how patients were allocated to MT or non-MT groups would enhance the transparency of the study.
- The results are clearly presented, showing significant increases in weight and albumin in the MT group compared to decreases in the non-MT group.
- The comparison of baseline NRI between the groups and the changes over the 12-week period are reported, supporting the conclusion that MT positively affects NRI.
- The manuscript effectively highlights the clinical implications of the findings, recommending personalized nutrition and routine follow-up based on NRI before MT initiation.
- However, the suggestion of routine NRI-based follow-up requires further justification. Future studies could explore the long-term effects of MT on NRI and clinical outcomes to validate this recommendation.
- The statistical analysis appears robust, with appropriate use of p-values to assess significance. However, additional details on the statistical methods employed (e.g., regression analysis, adjustment for confounders) would strengthen the validity of the findings.
- The conclusion is well-supported by the study results and underscores the importance of considering NRI in the management of CF patients.
- It would be beneficial to include future research directions or potential clinical implications to provide a more comprehensive conclusion.
Overall, the abstract presents a valuable contribution to the literature on CF management and provides evidence for the positive impact of MT on NRI.
Addressing minor points such as providing more context on NRI and detailing the statistical methods would enhance the clarity and robustness of the study.
Author Response
The manuscript "Effect of Modulator Therapies on Nutritional Risk Index in Adult Cystic Fibrosis Patients: A Prospective Cohort Study”, presents a prospective cohort study comparing the nutritional risk index (NRI) in adult cystic fibrosis (CF) patients receiving modulator therapies (MT) versus those receiving only non-modulator (conventional) therapies (non-MT). The study assesses weight gain and albumin levels over a 12-week period in both groups.
Thank you for your valuable comments.
Comments:
- The study design as a single-center prospective cohort study is appropriate for assessing the impact of MT on NRI.
Thank you
- The manuscript mentions a pragmatic design based on individual patient access to MT for 12 weeks, which could potentially introduce bias. Providing more details on how patients were allocated to MT or non-MT groups would enhance the transparency of the study.
These sentences have been added as you request:
As a result of the decision made by the responsible physicians of the patients based on the gene tests and the clinical condition of the patient, independent of the study and based on international treatment guidelines, some of the patients continued to receive conventional therapies, while in others modulator therapies were added. Therefore, the data obtained in the study completely reflect real world data, with the strengths and limitations of such a design.
- The results are clearly presented, showing significant increases in weight and albumin in the MT group compared to decreases in the non-MT group.
Thank you
- The comparison of baseline NRI between the groups and the changes over the 12-week period are reported, supporting the conclusion that MT positively affects NRI.
Thank you
- The manuscript effectively highlights the clinical implications of the findings, recommending personalized nutrition and routine follow-up based on NRI before MT initiation.
Thank you
- However, the suggestion of routine NRI-based follow-up requires further justification. Future studies could explore the long-term effects of MT on NRI and clinical outcomes to validate this recommendation.
Thank you for pointing this out. These sentences have been added to the conclusion section as you request:
However, the suggestion of routine NRI-based follow-up requires further justification. Future studies could investigate the long-term effects of MT on NRI and clinical outcomes to validate this recommendation.
- The statistical analysis appears robust, with appropriate use of p-values to assess significance. However, additional details on the statistical methods employed (e.g., regression analysis, adjustment for confounders) would strengthen the validity of the findings.
Thank you for your very important suggestions. After critical statistical analyses, these sentences have been added to the method and result sections:
Methods: These univariate analyses identified demographic and NRI variables with p-values below 0.20, and a logistic regression was performed to determine their effect on the likelihood of changing treatment type (modulator vs. non-modulator).
Results: The logistic regression model was statistically significant. The model explained 74% (Nagelkerke R2) of the variance in therapy (modulator vs. non-modulator) and correctly classified 86% of cases. However, there was no significant relationship between the de-mographic and NRI variables and the treatment type.
- The conclusion is well-supported by the study results and underscores the importance of considering NRI in the management of CF patients.
Thank you very much
- It would be beneficial to include future research directions or potential clinical implications to provide a more comprehensive conclusion.
As I mentioned above, these sentences were added to the conclusion:
However, the suggestion of routine NRI-based follow-up requires further justification. Future studies could investigate the long-term effects of MT on NRI and clinical outcomes to validate this recommendation.
Overall, the abstract presents a valuable contribution to the literature on CF management and provides evidence for the positive impact of MT on NRI.
Thank you
Addressing minor points such as providing more context on NRI and detailing the statistical methods would enhance the clarity and robustness of the study.
More context about NRI has been provided within the manuscript and statistical methods have been detailed with regression analyses.

Round 2
Reviewer 1 Report
Comments and Suggestions for Authors
No Further comments.